# Comprehensive Understanding of Foot Development in Children Using Capacitive Textile Sensors

**DOI:** 10.3390/s22239499

**Published:** 2022-12-05

**Authors:** Sarah De Guzman, Andrew Lowe, Cylie Williams, Anubha Kalra, Gautam Anand

**Affiliations:** 1School of Engineering, Computing and Mathematical Sciences, Auckland University of Technology, Auckland 1010, New Zealand; 2Faculty of Medicine, Nursing and Health Sciences, Monash University, Melbourne, VIC 3800, Australia

**Keywords:** foot length development, textile sensors, proximity capacitive sensing

## Abstract

Knowledge of foot growth can provide information on the occurrence of children’s growth spurts and an indication of the time to buy new shoes. Podiatrists still do not have enough evidence as to whether footwear influences the structural development of the feet and associated locomotor behaviours. Parents are only willing to buy an inexpensive brand, because children’s shoes are deemed expendable due to their rapid foot growth. Consumers are not fully aware of footwear literacy; thus, views of consumers on children’s shoes are left unchallenged. This study aims to embed knitted smart textile sensors in children’s shoes to sense the growth and development of a child’s feet—specifically foot length. Two prototype configurations were evaluated on 30 children, who each inserted their feet for ten seconds inside the instrumented shoes. Capacitance readings were related to the proximity of their toes to the sensor and validated against foot length and shoe size. A linear regression model of capacitance readings and foot length was developed. This regression model was found to be statistically significant (*p*-value = 0.01, standard error = 0.08). Results of this study indicate that knitted textile sensors can be implemented inside shoes to get a comprehensive understanding of foot development in children.

## 1. Introduction

Literature on foot morphology indicates that in the first 3 years of life, the foot grows approximately 2 mm in length per month. More detailed studies indicate different foot-growth rates for children ages one to five years old. Both genders grow at similar rates; however, boys’ feet are likely to grow one size longer and one size wider [1,2]. It is also indicated that the development of the child’s foot is strongly influenced by the point at which the child starts to stand and start walking. Most manufacturers create shoes that are miniature adult shoes. However, children and specifically toddlers do not have the same foot anatomy and characteristics as adult feet [3,4,5,6,7]. During infancy, there is no need for a shoe as they will not be walking. From the phase when they start to walk, from ages 1 to 2 years, the sole purpose of the shoe is protection from weather and the environment [8]. Therefore, at this age, shoes must be very soft and flexible to allow for freedom to move as though one were barefooted. Thick soles would impinge foot development as they add strain to the foot and ankle joints [9]. Podiatrists still do not have enough evidence as to whether footwear influences the structural development of the feet and associated locomotor behaviour [3,8,10]. Information on foot growth can provide important insights on topics such as the occurrence of a child’s growth spurt and an indication of when to buy new shoes [11]. Although foot-growth research is known to be one of the best indicators of a growth spurt and possibly various diseases, information regarding when parents buy new shoes and how their children’s feet grow inside these shoes is lacking [11]. This work describes the development and validation of in-shoe monitoring tools to observe children’s foot growth and development, including foot shape and structure (i.e., anthropometric parameters), and specifically foot length. This paper aims to study the technical feasibility of textile sensors in measuring foot length by creating and validating shoe foot-length measurement systems to alleviate the pain points of both parents and podiatrists. This was undertaken by developing two prototypes, grounded heel and non-grounded heel, with embedded textile sensors for capacitive proximity sensing of the foot. Capacitive sensing is based on measuring electrical output through coupling between conductive and dielectric mediums. The change in capacitance detected through these sensors is directly proportional to the changes in pressure applied to the foot. Hysteresis is one of the critical drawbacks of capacitive sensors employing polymers, as they tend to lose their water-retention properties over time; thus, in this study, we made use of textile sensors to avoid this problem.

Textile sensors are made of intertwining fibres (also known as yarns or thread) formed by knitting, weaving, felting, and crocheting yarns together [10,12]. The current trend of using smart textiles shows various potential applications along with several challenges. Many smart textile technologies have been proven in laboratory studies, but there are still uncertainties in mass production. An initial investigation by Harms, Amft, and Troster [13] showed positive trends in simulations predicting the behaviour of loose and partially fitting garments. 

## 2. Materials and Methods

In this study, the development and feasibility of textile sensors to better indicate the foot length of a child while the foot is still inside the shoe are discussed. There are two prototypes developed to test out its feasibility:Prototype 1 explains the process of selecting suitable sensing materials, sensing configurations, and electrical components. The materials and sensing configuration are evaluated quantitatively and qualitatively.Prototype 2 is a design iteration of the previous prototype. It uses the selected materials from Prototype 1 and refined design configuration. The effectiveness, reliability, and feasibility of this prototype are tested and evaluated with children using a wireless electronics module.

Both prototypes follow the proximity-capacitive-sensing principle. The desired sensing electrodes are made of conductive textiles located at the toe cap of the shoe. These are connected by coaxial cables to the Arduino 101 and MPR121 capacitive sensing chips. These two chips make up the embedded electronics unit to enable reading the fabric sensor’s output. The output is then passed on to a data logger for further analysis and evaluation. In Sensor Prototype 1, the Arduino 101 is connected to a PC to enable data logging. In Sensor Prototype 2, the Arduino 101 is replaced by a smaller electronic Pycom unit to enable wireless communication from the electronics unit to an Android mobile phone. Wireless communication is implemented as it created a safer testing environment during the evaluation of the prototype with children.

### 2.1. Sensor Prototype 1

The Sensor Prototype 1 design requirements are broken up into two categories: mechanical aspects and electrical/electronic components and software. For the mechanical aspects, the sensor layout and materials and knitting techniques are included. Physical connections from the knitted conductive textiles sensors and other e-textile components and the interfacing of MPR121 (capacitive sensing chip) to Arduino constitute the electrical/electronic components and software. These prototype components and test procedures are explained in the following subsections.

#### 2.1.1. Sensor Placement, Size, and Layer Layout

To create Sensor Prototype 1, conductive textile materials from Auckland University of Technology (AUT) Textile and Design Lab were obtained, and samples were knitted by the knitting technician for a specific size of toe cap. The toe-cap dimensions were obtained from the Bobux design database (Bobux is a New Zealand children’s shoe company that started creating soft sole shoes for infants in 1991). Two different fabric sensors were acquired, and these were installed inside two different shoes of the same size. The fabric sensors are connected with a coaxial cable to reduce outside interference/noise in the readings. The sensors were read initially by the Agilent E4980A LCR meter (Keysight Technologies, Santa Rosa, CA, USA, Agilent E4980A) and again by the Arduino 101 with an MPR121 capacitive sensing chip. Arduino 101 was connected to a PC and data were logged through Arduino’s built-in serial monitor. The system layout is depicted in Figure 1.

The evaluation of conductive textiles to select suitable sensor materials was carried out by inserting different-sized phantom feet. The readings were obtained using an LCR meter and these results were compared and graphed. The criterion for choosing a suitable sensor material is the linearity of the sensor results.

Normally, foot length is measured outside the shoes. In typical foot-measuring devices, the heel of the child needs to coincide with the curved part of the device, and a horizontal bar is moved until it touches the longest part of the toe. The number that this bar stops at indicates the size of the child’s foot. Ideally, the heel sits at the heel counter. This is designed so that it can hold the heel in position, giving the foot more stability during variable movements, together with the whole shoe upper, which firmly holds the foot inside the shoe. This creates a basis for where to place the sensors. The heel will not experience much change, because the heel will always stay in the heel counter. Therefore, the toe area is an ideal place to put the sensors, as this would see the most change in length. This area is also ideal as layers can be easily fitted and concealed. In shoe manufacturing, polyester material is sandwiched between the inner layer of the shoe and the protective outer layer of the shoe as depicted in Figure 2. This polyester material (called a toe puff) hardens the toe area, which keeps the shoe in shape and adds more protection from scuffing during child’s play.

Since the sensing technology is based on proximity sensing, there needs to be a conductive area that would serve as the electrode/sensing area. This is achieved using knitted conductive threads placed in the toe-cap area of an existing Bobux shoe. The sensor shape is based on the toe-cap template provided by Bobux shown in Figure 3. The print area indicates the sizing limit of the sensor shape.

Apart from the sensor, there must also be an insulation layer, to support the sensor in place, and a shield layer to protect the sensor from false touches that can occur outside the shoe. This is an unwanted signal, which comes from an external influence, not the shoe wearer. The detection of the foot should only happen from the inside of the foot. The shield layer is another conductive fabric that helps decrease the interference from the outside of the shoe. The sensor, insulation, and shield make up the three layers of the toe-cap area of the shoe. The textile sensor transmits data to the electronics unit using the black and red wires shown in Figure 4. The black wire is connected between the shield and the ground (GND) pin of the electronics unit; the red wire is connected between the sensor and the signal pin of the electronics unit.

Additionally, the red wire (sensor/data wire) needs to also be shielded so as not to transmit false touches. This is achieved using a coaxial cable, where the innermost wire of the coaxial cable is connected to the sensor layer, and the outermost loose-braided copper wires are connected to the shield layer.

#### 2.1.2. Sensor Materials

There were two conductive threads available in AUT Textile and Design Lab with different conductivities, as shown in Figure 5:

(a) Silver thread—2/117 dtex 99% silver-plated nylon, linear resistance 3 kΩ/m (Statex Productions and Vertriebs GMBH, 2010)

(b) Stainless steel thread—2/50 Nm, 80% polyester and 20% stainless steel, linear resistance 530 kΩ/m (Ehrmann, Heimlich, Brucken, Weber, and Haug, 2014)

Both threads were plain knitted as a sheet with ribbed edges using a SIG 123 SV 14 (Shima Seiki, Castle Donington, UK) gauge knitting machine. The sheets were laser cut with the toe-cap template. This is to prevent fraying around the sensor edges. The laser-cut sensors were shaped around the toe-cap area of the shoe last using masking tape. This is undertaken so that the knit would create a solid shape, rendering it easy to attach inside the shoe as shown in Figure 6.

Figure 5B shows the different layers of sensor, insulation, and shielding components of Sensor Prototype 1. Sensor components shown include stainless steel and silver knitted sensors. These two materials were evaluated for their effectiveness, and the results presented in Section 3. Insulation components are the layers of the shoe, which includes (top to bottom) the sock liner, toe hardener/toe puff, and synthetic leather. The shielding component shown is copper taffeta. The sensor and shielding components are separated by the insulation components.

Sensor Prototype 1 uses conventional braided shielded copper wires. This type of wire contains a conductive inner core, usually copper insulated with PVC. This is surrounded by another set of copper wires braided around it, used for shielding. The innermost wire is shielded so that less software filtering is required to isolate the data needing to be read. This removes the interference that can occur when it is installed inside the shoes. This interference can occur when the skin from other parts of the foot touches the wire, such as when a child inserts their foot inside the shoe. This sliding motion along the wire can cause unwanted readings.

#### 2.1.3. Electrical/Electronic Components and Software

The electrical/electronic components will read the capacitance signals from the textile sensor and connections. There are two components used for the capacitance reading and for the acquisition of data from the sensor to the PC: the MPR121 and Arduino 101, as shown in Figure 7.

The breakout board from Adafruit Industries includes an NXP semiconductor capacitive sensing chip called the MPR121. This can have up to 12 electrode inputs. One input electrode is used in foot-length measurement. This board was chosen due to its small form factor and capacitive sensing range from 10 pF to 2000 pF with a resolution of up to 0.01 pF. This range is desirable due to the unknowns of the capacitance of the fabric sensors.

This capacitive sensing module uses a CTMU to determine capacitance changes. This involves the measurement of voltage, as the system assigns a known current on the sensor. The circuit then examines the shift in the voltage while constant current is being applied over a specified amount of time.

MPR121 can be set up as a touch and proximity sensor. In this case, it is set up as a proximity sensor. This microcontroller built into the Adafruit board is already wired so that it can step down higher voltages for up to 5V, as it only needs 3.3 V. MPR121 is also an I2C compliant device; it has a configurable I2C address that can be used to change register values to program the microcontroller as wanted. MPR121 is interfaced with an Arduino 101 module, this allows communication and programming in Arduino IDE without having to change the source code. Communication is via I2C, and the information from the sensing module is accessed through registers via serial data.

##### Code Implementation

For MPR121 to communicate and read the correct capacitance values, an algorithm is followed as recommended by the MPR121 datasheet. Open-source code and libraries that are readily available online were used with alterations to fit the paper’s objective. Pseudocode is described in Figure 8, detailing the functions used in the program.

The algorithm components are briefly explained as follows:Include header files and define device registers: As MPR121 is an external sensing chip, libraries and header files (i.e., *#include* “*Adafruit_MPR121.h*”) are included so that variables inside this file can be easily accessed. After introducing the header files, the device addresses must be defined so that Arduino knows which information to access.Define constant, reading, and configuration registers: Once the device is addressed and found, configuration registers are defined so that their values can be changed in the setup functions. There are corresponding values for these configuration registers and only some registers are selected, as not all the electrodes are being used. Some registers are not configured. However, they are introduced here as they will be read later, in the loop function. These registers change based on the changes made to the configuration register. The registers that need to be read are *ELEC_Current*, for reading the electrode’s current; *ELEC_Time*, the register for the electrode’s charge time; and ADC Registers, *ADCLSB*, and *ADCMSB*, for reading the electrode’s voltage.Setup function: the baud rate is defined to know how quickly the microcontroller samples and how often it obtains information from the serial port. It is set to read at the fastest rate, at 115,200 baud. The introduced configuration registers are changed to their corresponding values to perform as it needs. To set these registers to their new value, the device itself must be soft reset (*SOFT RESET*). This resets the registers into their default values. After this, the MPR121 is at stop mode (*ELEC_CFG* bit assignment is 0b10000000), so the remaining registers, such as the auto configuration registers (*AUTOCONFIG0*) and voltage operation limits (*UPLIMIT*, *LO LIMIT*, *TARGETLIMIT*), may be changed. Once registers are set, the MPR121 is changed to run mode so that these settings can be applied and used. To enable run mode, the electrode-configuration register-bit assignment changes to enable electrode detection and proximity sensing at electrode 0.Loop function: In this function, certain registers are read to enable the calculation of capacitance. As mentioned earlier, MPR121 operates using CTMU, which means that it needs current, voltage, and time to calculate capacitance. These can be read from reading registers defined earlier (*ELEC_Current*, *ELEC_Time*, *ADCMSB*, *and ADCLSB*). The current register can be read as is and does not need any other conversions. The time register contains two values, each corresponding to the charge time for that electrode. The electrode used for Sensor Prototype 1 is Electrode 0, the time for this register is accessed on the *ELEC_Time* register; however, this contains time values for electrodes 0 and 1.

#### 2.1.4. Sensor-Material Evaluation and Code-Verification Testing

This is undertaken to evaluate whether the capacitance reading truly increases when the toe area gets closer to the sensor, which is laid out in a non-planar pattern. Additionally, this is to evaluate if the programmed capacitive sensing circuit displays a similar output as the LCR meter, which is a more reliable source of readings. In addition, an evaluation is needed to compare the silver knitted threads and stainless steel threads in terms of sensing proximity.

To test whether capacitance truly increases when the toe gets closer to the sensor, 3D-printed shoe lasts were covered in aluminium conductive tape. This is to replicate the conductive characteristics of the human foot and will be referred to as the foot phantom. Although aluminium may not have the same conductivity and electrical behaviour as the human foot, these were sufficient to undertake the tests carried out for Sensor Prototype 1. These lasts are of varied sizes; the range used was sizes 20–23 with equivalent lengths between 13.5 cm (Size 20)–15.5 cm (Size 23) as shown in Figure 9A. The test setup is shown in Figure 9B.

To assess whether the Arduino was programmed correctly, the textile sensors inside the shoes were connected to both the Agilent E4980A LCR meter and the Arduino and the readings were observed using the Arduino serial data window and recorded.

Finally, the results recorded from both materials were then evaluated. A criterion used to select material is that the capacitance should increase linearly as the foot approaches the textile sensor.

The results and discussion of Prototype 1 are found in Section 3.

### 2.2. Sensor Prototype 2

The objective of this prototype is to create a full textile and connection system so that it can conform to any shape and be more robust to flexing. An enclosure is created to protect the electronics unit during testing with real human feet. In addition, the prototype attempts to increase the sensitivity of the fabric sensors, so that they can read a larger range of values over a small distance.

Based on Prototype 1, the maximum sensing range corresponds to a foot phantom 2 cm away from the sensor. This corresponded to the insole information provided in Figure 10a. This refers to the 2 cm mark, which matches a foot from size 20 (13.5 cm) to size 23 (15.5 cm). From this information, a new striped pattern, Figure 10b, was devised to create a sensor that can sense a larger range, while also improving the structure of the knit integration with other threads to be implemented as a sock liner.

The striped pattern was drawn in Solidworks and corresponds to a size-23 shoe insole and toe cap. The dimensions of the pattern are shown in Figure 11. The thickness of the conductive knits and gaps are determined by the minimum number of courses (rows) the knitting machine can achieve. A SIG 123 SV 14-gauge knitting machine was used, which can do a minimum of two courses.

Two courses are equivalent to 2 mm thickness; the conductive traces are knitted with four courses (wc) and the non-conductive thread gaps (wnc) are knitted with two courses.

Before implementing this design with the knitting technician, the striped pattern was first implemented as a cut-out of copper stripes. The template was printed on a one-to-one scale, and the copper tape was cut out to the same scale. The stripe running across the conductive widths was replicated in the mock-up by connecting a pure stainless steel conductive thread with a single header pin. Figure 12 contains an amber-coloured tape, called polyimide tape, which provides insulation and electrical isolation from the surrounding electronics and human touch.

The mock-up was plugged into the Arduino with an MPR121 unit, to see whether capacitance increased whenever a copper stripe is covered. This was undertaken by covering each stripe with each size of the foot phantom (size 20–23) and observing whether there was an incremental change whenever a copper stripe is covered. The results were read and logged using the Arduino serial monitor built into the PC.

The graph depicted in Figure 12 shows an incremental increase when each stripe is covered by a size 20–23 foot phantom. Although this proved that the striped design works, this test is not an overall indication of the final design, as the final prototype must be made of fabric and not paper and copper tape.

### 2.3. Sensor Manufacturing—Sensor Fitment Iterations

Different iterations of sensors, mainly three versions, are assembled and visually evaluated. Version 1 is similar to the copper stripes but replaced with knitted 80 PES/20 SS conductive thread. For Version 2, the textile sensor was made and backed up with a polyester thread (white), to hold up the conductive stripes evenly. The stripes are also more complete, as it does not have to be hand-sewn together.

Version 3 was knitted as a sock. This eliminates the handwork of joining the upper and the sole-fabric sensor. The type of knitting used for the samples is tubular knitting using Shima Seiki Whole Garment technology. This technique allows for two different materials to be knitted together. It incorporates non-conductive polyester threads in between the conductive threads, allowing for a better mechanical structure and preventing the conductive rows from touching each other. This also allows for a seamless connection along the stripes, ensuring each row is connected as one round stripe. The main connection from the fabric sensor is performed the same way in Version 2, where an insulated stainless-steel thread was stripped and sewn onto the sensor. The knitted sensor sock was coated in PVA glue while it was inside a last. The glue was cured and dried inside an oven at 100 °C until dry to touch. This is to keep the sensor in shape and the insulated conductive thread connection in place, for easy installation inside the shoes.

### 2.4. Final Sensor Assemblies

Version 3 was chosen for manufacturing the final assemblies of the sock sensors because it achieved a full textile connection and the seamless integration of the knitted sensor from upper to sole. There were two knitted sock-sensor samples created for the test, characterized as grounded heel (GH) and non-grounded heel (NGH), as shown in Figure 13. Both samples were knitted with a combination of non-conductive polyester thread and 80/20 PET/SS conductive thread. GH contained a conductive knit patch on the heel and NGH did not. GH and NGH samples had the same knit layout on the toe area.

The sensors were installed inside Bobux shoes. The GH sensor contained an additional ground plane at the heel. This was connected to the shield layer on the outer upper of the shoe (copper taffeta). For both GH and NGH, insulated stainless steel thread was used as the sensor wire, carrying data to the electronics unit. A non-insulated stainless steel conductive thread is wrapped around this thread, to replicate the shielding effect of a coaxial cable, preventing noise from the environment.

Figure 14 shows the physical connections of the shoe with the GH sensor. The black cloth contains connections from the copper taffeta and the extra ground plane on the heel. The connections shown in Figure 14(right) are connected and crimped to a blue female FCI clincher. This was used as it gives the best contact for the traces while ensuring reliable contact between the conductive threads and the electronics unit.

### 2.5. Pycom Expansion Board and WiPy 2.0

A wireless system was implemented so that the device can be tested on children’s feet. This is undertaken to avoid any hazard that may occur during testing, such as a broken connection between the Arduino and the PC due to accidental disconnection while the child is walking.

The Arduino is replaced by two modules, a Pycom Expansion Board and WiPy 2.0. The Pycom Expansion Board is used because of its easy plug-and-play configuration, which allows for external microcontroller boards to be plugged in. In this situation, the MPR121 is still used and is plugged into the exposed female pin headers using a custom-made PCB.

The Pycom expansion board includes a mini–USB plug so that it can be easily programmed and deliver power using an external power source such as readily available power banks. The expansion board contains a step-down voltage regulator, so that when a higher voltage is plugged in, it will only take what is needed to prevent overvoltage (Pycom Ltd., Guildford, UK, 2017).

The WiPy 2.0 is a separate microcontroller that is attached by inserting the headers on the inner part of the Pycom Expansion Board. This microcontroller features Bluetooth low energy (BLE), Bluetooth classic, and wi-fi. This is advantageous, as it can be integrated with older Android devices that do not support BLE. It can also communicate over wi-fi, therefore eliminating the need for a wired connection to log and acquire data. This is compatible with MPR121, as it can communicate with the I2C bus protocol (Pycom Ltd., 2017).

An external custom-made PCB shield was manufactured to decrease the number of wires connecting the MPR121 to the Pycom Expansion Board headers, as shown in Figure 15.

The Arduino 101 is replaced by the Pycom Expansion Board and WiPy 2.0. Although both WiPy 2.0 and Arduino 101 support a Bluetooth low energy module, the WiPy 2.0 also had Bluetooth classic. This version had more software support in terms of open-source code. It was also more compatible with older Android phones, as they only work with Bluetooth classic.

An Android app was made using Android Studio to allow for seamless data collection, creating a wireless data-logging system. This meant that the electronics unit was capable of operating while it was strapped on a child, without worrying about physical disconnections. The code of MPR121 implemented in Arduino in Prototype 1 was also implemented in WiPy 2.0; this provided the same reading conditions with a different electronics unit.

A hardware enclosure as shown in Figure 16 and Figure 17 was created to prevent physical disconnections between the circuit and the shoe. The hardware enclosure also prevents possible electric shocks that may come from low-voltage electronics while they are being tested on children. The design of the enclosure was based on an original Pycom enclosure provided on purchase.

#### 2.5.1. Overall Connection and Assembly of Shoes and to Electronics Unit

In Prototype 1, the overall system layout considered the use of Arduino. In this section, Pycom with MPR121 is implemented. The overall physical connections with pin assignments are laid out in Figure 18.

The grounded heel sensor and non-grounded heel sensor connected to Pycom and MPR121 are shown in Figure 19.

#### 2.5.2. Evaluation with Children

A study on children was undertaken to prove the effectiveness and reliability of the textile sensors. Ethics approval was granted by the Auckland University of Technology Ethics Committee (AUTEC). All children in this study were recruited from the general community using online platforms and email recruitment from the AUT Early Childhood Centre and the existing Bobux customer database. In-person recruitment also occurred using posters put up in the Bobux outlet store. Parents and/or guardians gave written consent and children assented to participate. During the test held in the Bobux retail store, the parents and caregivers needed to be present to give them familiarity with the environment. The shoe prototypes were tested on 30 children, and 14 datasets were taken from that.

The inclusion criteria put in place for this work were that only developing children aged 11–36 months old would be included, and parents would be asked if their children had any known health conditions that could have an impact on walking. These include genetic conditions that change walking, neuromuscular conditions that are exhibited in tiptoe walking, or orthopaedic conditions that result in foot-structure change.

Necessary exclusion criteria were: only children that can stand and walk properly could participate, and children’s foot length needed to be under 150 mm, as the sensors are instrumented inside a size 23 Bobux shoe, which has a foot length of 150 mm.

The test procedure called for the participants/children to be in a static (standing) position, with their heels at the back of the shoe, and with socks worn on both feet. During the test, the children’s guardians were asked to count from 1 to 10 after physical measurements—i.e., after foot length and foot circumference were measured. During the test, other parameters were recorded, such as Bobux shoe size (according to the Bobux shoe ruler), recorded by placing the child’s heel at the heel curve of the Bobux shoe ruler, and the slider on the opposite end being slid until it touched the longest point of the foot, as shown in Figure 20.

Foot length was also measured using a tape measure, in which a tape measure was placed at the back of the Bobux shoe ruler. This indicated the foot length in centimetres. Foot circumference was measured using a tape measure (as shown in Figure 21), in which the tape measure was wrapped around the widest part of the foot while the participant was standing up.

These were recorded to create a comparison between capacitance readings, foot length, indicative Bobux shoe size, and possibly foot volume.

## 3. Results and Discussion

This section includes the results of both Sensor Prototype 1 and Prototype 2.

### 3.1. Sensor Prototype 1 Results and Evaluation

Tests were performed to see whether the programmed MPR121 interfaced with Arduino 101 read a similar output as the LCR meter (Keysight Technologies, Santa Rosa, CA, USA, Agilent E4980A), which is a more reliable source of measuring capacitance. Table 1 displays the capacitance readings from both knitted textile sensors, where the readings were derived from an LCR meter and the Arduino.

Figure 22 compares the capacitance readings of the silver samples, and Figure 23 compares the capacitance readings of the stainless-steel samples.

Both samples in Figure 22 behaved the same way, regardless of which measuring system was used, as the Arduino readings followed the same pattern as those from the LCR meter. However, the scaling between the two readings is different, because the readings did not have the same starting point and there was a large offset between two readings. There is also an irregularity, where the reading of Arduino at 1.5 cm is higher compared to the readings at 1 cm.

Both samples in Figure 23 also behaved similarly regardless of the measuring system. As the phantom foot approached closer to the sensor, the capacitance increased. The trends for both Arduino and LCR meter are similar; however, between the 2 cm and 1 cm mark, they have two different slopes. The Arduino detected a much larger change on the last centimetre compared to the LCR meter, resulting in a much steeper gradient and dramatic change as the foot came much closer to the sensor.

Although the readings from the Arduino are not the same as those from the LCR meter, both measurements followed the same trend/pattern. This is a good indication that the Arduino interfaced with MPR121 is good enough to measure the capacitance changes.

From the results in Table 1, Figure 22 and Figure 23, as the foot phantom comes closer to the fabric sensor, the capacitance increases. This occurs regardless of what material it is made of or the equipment it is being measured with. All the knitted sensors can sense up to 2 cm away, as the graph starts to change its slope when the last reaches this part.

From the material evaluation point of view, as observed in Figure 24 below, the polyester/stainless steel (ss) sample (green line) showed a steeper increase in capacitance compared to the silver sample (blue line).

The SS sample also showed a more distinct increase when the toe last drew closer to the fabric sensor, whereas the silver sample exhibited an irregularity when it detected capacitance at the 1.5 cm mark. This may be because the sensor was not sitting flat in this area. As this sensor is shaped around a shoe last, there are areas in the sensor that may have more conductive material and so it bunched together, creating a thicker conductive plate. The more conductive material in this area means that it could have picked up more capacitance compared to an area before or after this. The bunching of sensor material is also possible as the knit material of the silver sample felt softer and more limp compared to the SS sample. The texture of the knit could have affected the readings when the lasts were inserted. The last could have shifted the sensor, creating an irregularity in the reading.

From Table 1, comparing the Arduino results, the two samples garnered two different capacitance ranges. The SS samples’ readings ranged from 26.3 to 76 pF, which corresponds to a capacitance range of 50 pF over 6 cm, whereas the silver samples’ varied between 53.19 to 80.5 pF—hence, a capacitance range of 27 pF over 6 cm. From these results, the sensitivity of the SS sample is greater (8.3 pF/cm) than that of the silver samples (4.5 pF/cm). This indicates that the SS sample has better sensitivity, which is more advantageous than the silver sample. A higher sensitivity means that the sensor can have more resolution in readings. This is suitable when measuring foot length because, especially from ages between 11 months old to 36 months old, the foot-length development is rapid. Therefore, a more sensitive sensor can pick up minor changes.

### 3.2. Sensor Prototype 2 Results and Evaluation

#### 3.2.1. Foot-Length and Capacitance—Statistical Analysis

Referring to Table 2, the GH shoe-sensor results possessed a statistically significant result (*p* < 0.05) and a steeper relationship compared to those for the NGH (*p* = 0.2). The standard error (SE) of the GH was smaller, statistically indicating that it is unlikely for the true gradient to be negative.

#### 3.2.2. Foot-Length and Capacitance Relationship Results

For these results, the circumference of the foot was taken from the widest part of the foot. An assumption was made that the volume of the foot has approximately the section of an ellipse. The volume of the foot is assumed to have a cylindrical ellipse shape; the semi-minor axis is assumed to be one-fourth the semi-major axis.

Referring to Table 3, the GH shoe-sensor results possessed a statistically significant result (*p* < 0.05) and a steeper gradient relationship compared to NGH (*p* = 0.06). The standard error (SE) of GH is smaller, statistically indicating that it is unlikely for the true gradient to be zero or negative.

#### 3.2.3. Prototype 2 Evaluation

The results shown in Figure 25, Figure 26 and Figure 27 showed the sensors are functioning when real human feet with socks on are inside the shoes. Both GH and NGH showed the feasibility of the textile sensors measuring foot length. This was indicated by the positive gradients in Figure 25, which mean that, as longer feet are inserted inside the shoe, the capacitance readings increase.

In Figure 25, *ygh* showed a steeper gradient and a larger r^2^ value compared to *yngh*. This may be due to the extra conductive pad in the GH sensor creating an effect that increases the sensitivity of the sensor. This extra conductive pad on the heel decreased the parasitic noise from the background. The result for *ygh* in Table 3 also showed statistical significance, as the *p*-value is less than 0.05 (*p* = 0.01), whereas for *yngh* it is greater than 0.05 (*p* = 0.2)—hence, the GH shows a more credible result compared to the NGH sensor.

Additionally, both GH and NGH sensors showed feasibility in measuring foot volume. This can indicate how much of the human foot is expanding inside the shoe. This is indicated by the positive gradients in Figure 27, as a larger volume indicates an increase in capacitance. However, a similar effect was presented where *yvgh* showed a steeper gradient and a larger r^2^ value compared to *yvngh* in the *p*-values in this linear regression model. Table 3 depicted lower values compared to the foot length and capacitance in Table 2. Even though the values are lower for the GH shoe, *yvgh* showed a more statistically significant result due to its *p*-value being 0.008, whereas for the NGH shoe, the *yvngh p*-value was 0.06. Again, this could have been the effect of the extra conductive pad on the heel of the GH shoe sensor.

The textile sensors appear feasible for sensing the foot length, especially the GH shoe sensor. However, the r^2^ value of the GH shoe is less than 50% when capacitance is compared against foot length and foot value. This can mean that the linear regression model may not be a suitable model for characterizing the textile sensors. Although the *p*-values showed credibility and proved the concept, more information, such as more variability in the foot lengths, is needed to create a conclusive and more definite capacitance-to-foot-length model.

## 4. Conclusions

To conclude, there were two aspects to this work. Sensor Prototype 1 tested the feasibility of textile sensors and Sensor Prototype 2 further improved the design of Sensor Prototype 1. Both sensor prototypes went through an iterative design process to create final prototypes, and these were tested and validated throughout the design process for further verification.

Sensor Prototype 1 showed that the physical presence of a phantom foot increases the capacitance reading of the fabric sensor. This was proved by reading capacitance using the gold standard measurement, an LCR meter (Keysight Technologies, Santa Rosa, CA, USA, Agilent E4980A). A smaller capacitance reading system was then implemented using Arduino and MPR121, to enable portable capacitance logging from the fabric sensor. The capacitance readings from both measurement systems provided comparable results. An issue found in this design iteration was that it was not sensitive enough to measure incremental changes. This means that the reading from a phantom foot 2 cm away from the sensor was similar to that of one 0.5 cm away from it.

A different sensor design was implemented in Sensor Prototype 2, to increase the sensitivity of the fabric sensors. Two sensor configurations were tested, to see which design would work better. In vivo validation was performed to examine whether a child’s foot would change the capacitance the same way a phantom foot would in the GH and NGH shoe. During this testing, the child’s foot length and foot circumference were measured with a measuring tape. The foot length and the capacitance reading were plotted in a graph for the GH and NGH shoe prototypes. The GH shoe indicated better performance compared to the NGH shoe. This was indicated by the difference in slopes and difference in standard errors, using the information analysed from linear regression. The *p*-value also indicated that the placement of an extra conductive pad in the heel part (GH shoe) increased the sensitivity of readings.

The positive increase in slope and small standard of error in the GH shoe indicated that textile sensors can measure a child’s foot length.

## Figures and Tables

**Figure 1 sensors-22-09499-f001:**
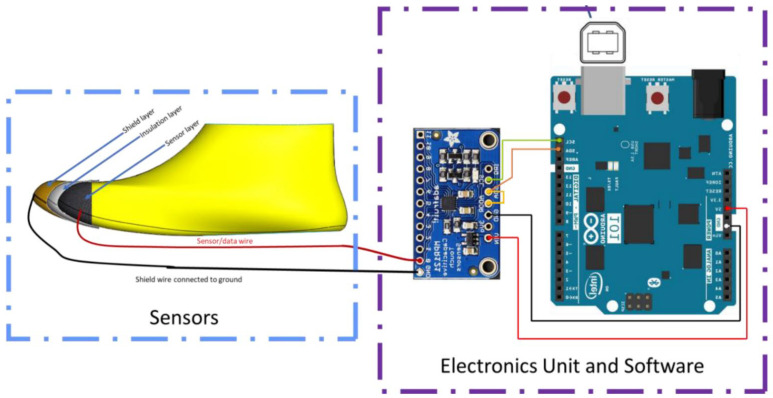
System layout for Sensor Prototype 1.

**Figure 2 sensors-22-09499-f002:**
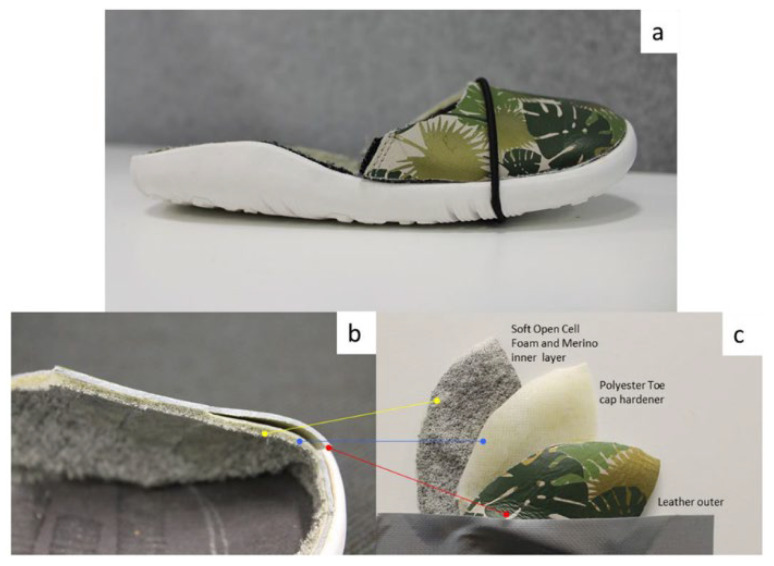
(**a**) Example of Bobux shoe with midsole removed; (**b**,**c**) Upper of the toe area showing the cross-section and layers of the shoe.

**Figure 3 sensors-22-09499-f003:**
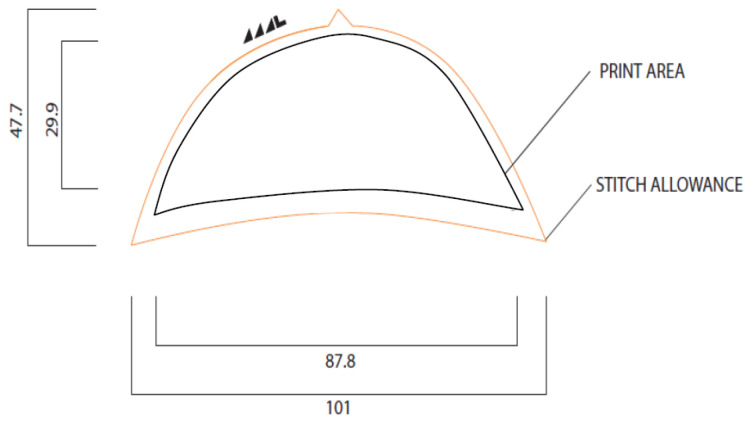
Toe-cap template of a Bobux shoe.

**Figure 4 sensors-22-09499-f004:**
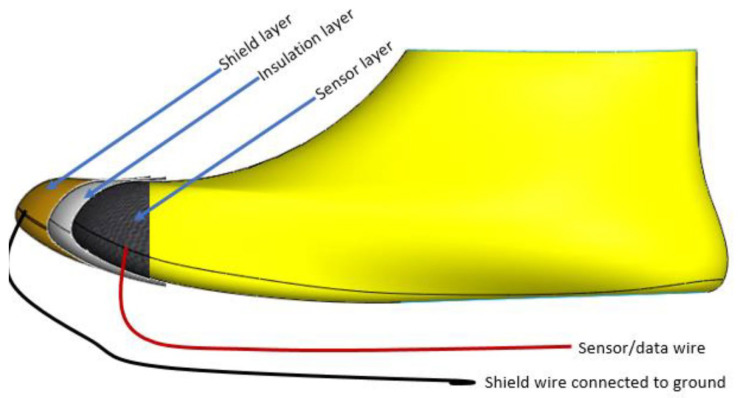
Sensor layer and wire layout.

**Figure 5 sensors-22-09499-f005:**
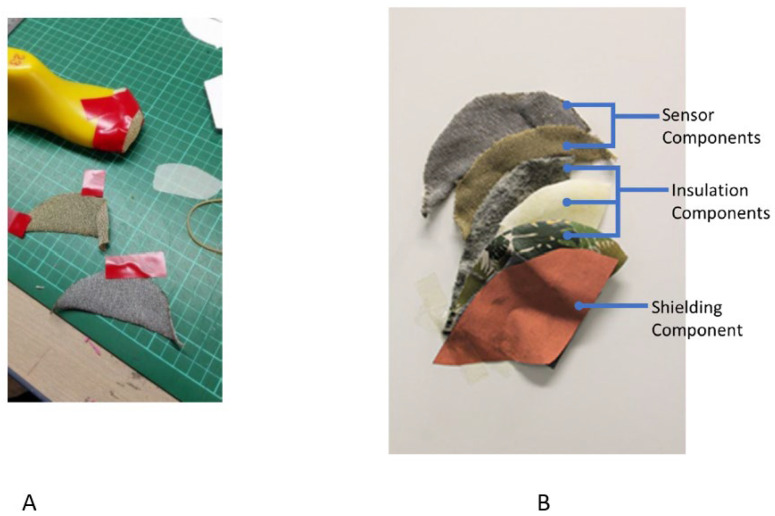
(**A**): Samples cut to shape using toe-cap template with (i) 99% silver thread sample, (ii) 80/20% polyester/stainless steel sample, (**B**): different layers of Sensor Prototype 1, showing the sensor, insulation, and shielding components.

**Figure 6 sensors-22-09499-f006:**
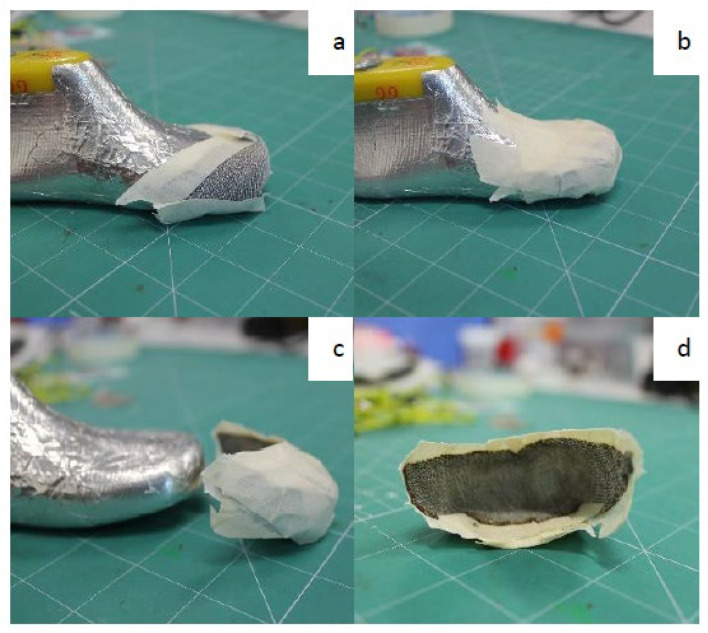
Example of the fabric sensors installed using 80/20 PES/SS sample. (**a**–**c**) low tack masking tape is used to shape the fabric sensor insulating the other layer. (**d**) completed sensor to go inside the toe area of the shoe.

**Figure 7 sensors-22-09499-f007:**
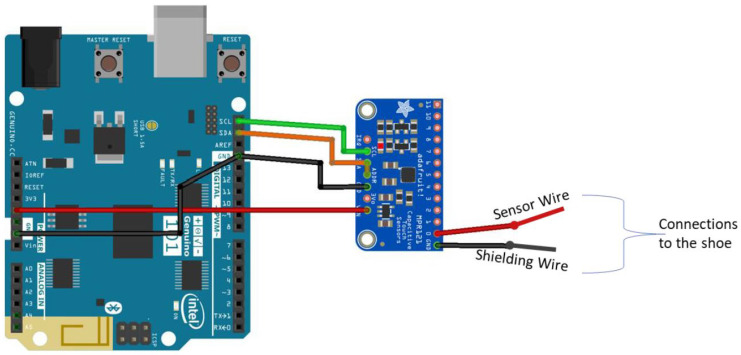
Connections between Arduino 101 and Adafruit MPR121.

**Figure 8 sensors-22-09499-f008:**
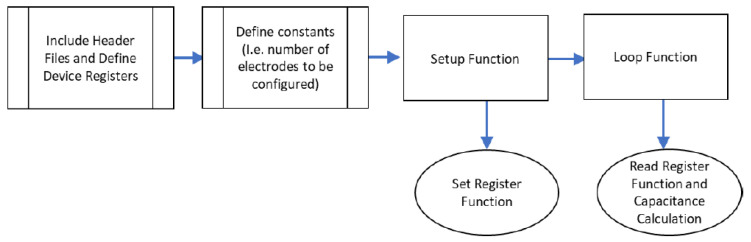
Pseudocode of the Arduino algorithm.

**Figure 9 sensors-22-09499-f009:**
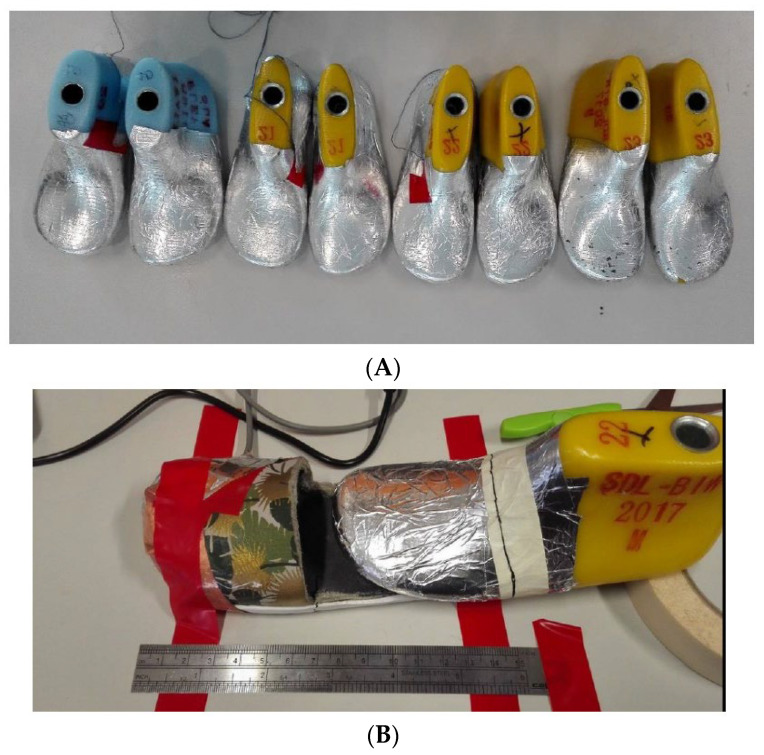
(**A**) Foot phantom, showing different sizes (showing sizes 20–23, left to right), (**B**) Test setup.

**Figure 10 sensors-22-09499-f010:**
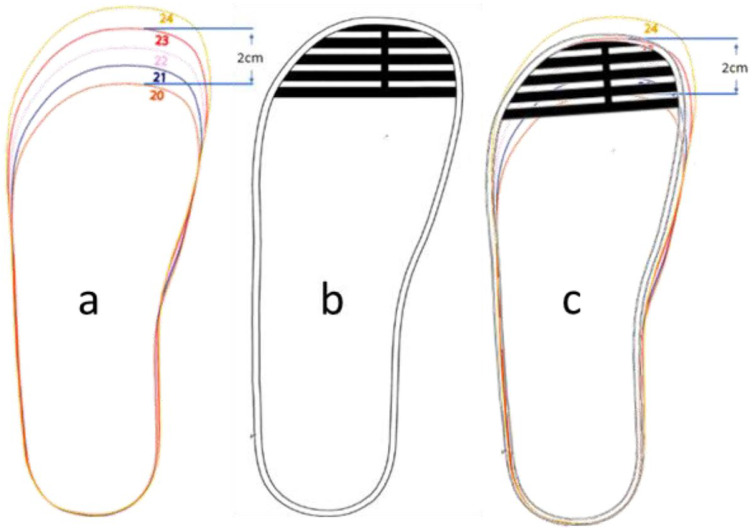
(**a**) Insole information from Bobux showing the 2 cm mark. (**b**) Sensor pattern segmented but connected in the middle. (**c**) Sensor pattern overlapped with insole information.

**Figure 11 sensors-22-09499-f011:**
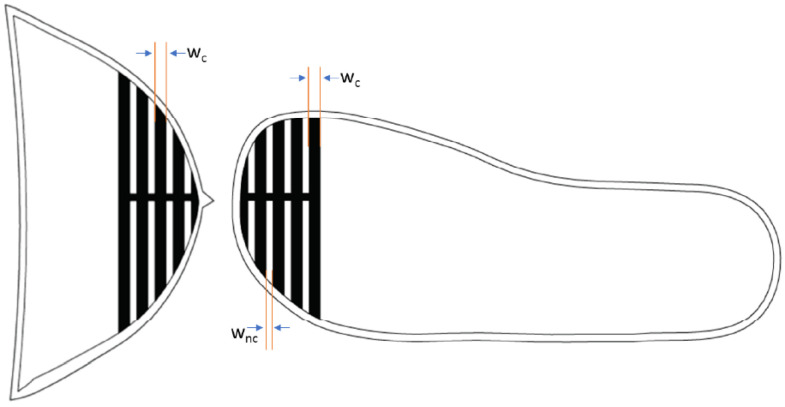
Striped pattern drawn on size-23 toe cap (**right**) and insole (**left**); w_c_ = width of conductive knit, and w_nc_ = width of non-conductive knit.

**Figure 12 sensors-22-09499-f012:**
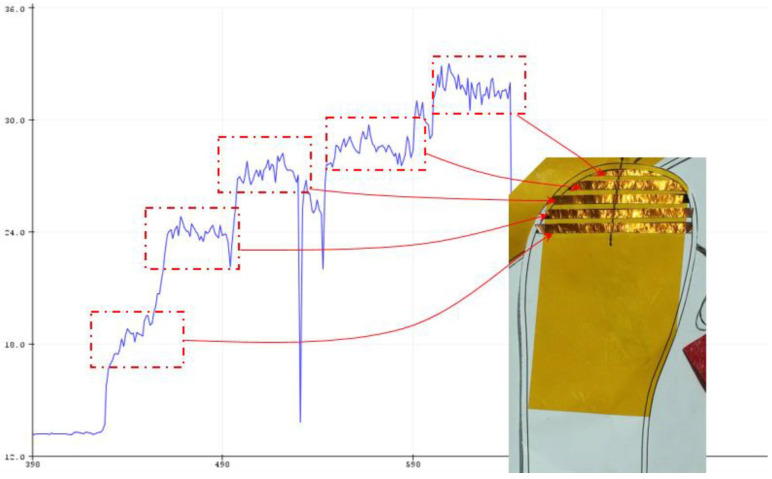
Incremental increase when lasts covered each copper stripe.

**Figure 13 sensors-22-09499-f013:**
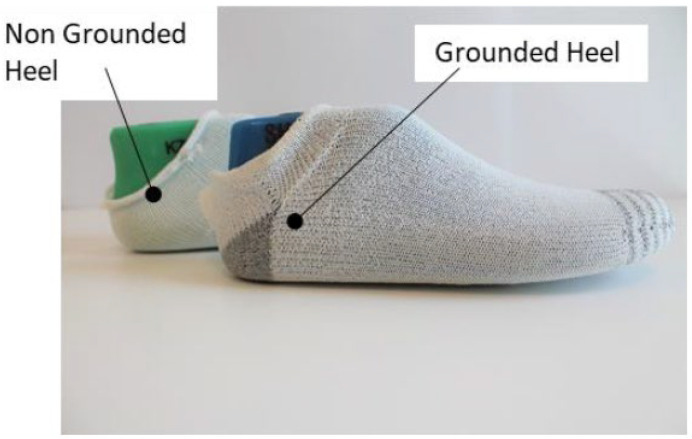
Two samples showing non-grounded heel (NGH) and grounded heel (GH) samples.

**Figure 14 sensors-22-09499-f014:**
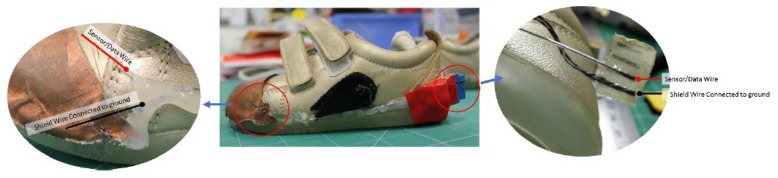
(**Left**) Connections from the sensor and shield layer on the toe cap; (**Middle**) Overview of the grounded heel shoe; (**Right**) Connections of wires attached to a *Z*-axis tape.

**Figure 15 sensors-22-09499-f015:**
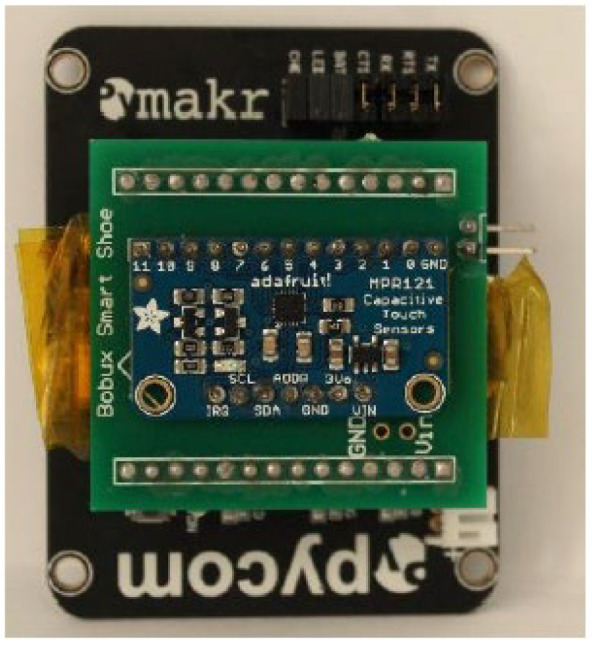
PCB shield assembled on top of the Pycom Expansion Board.

**Figure 16 sensors-22-09499-f016:**
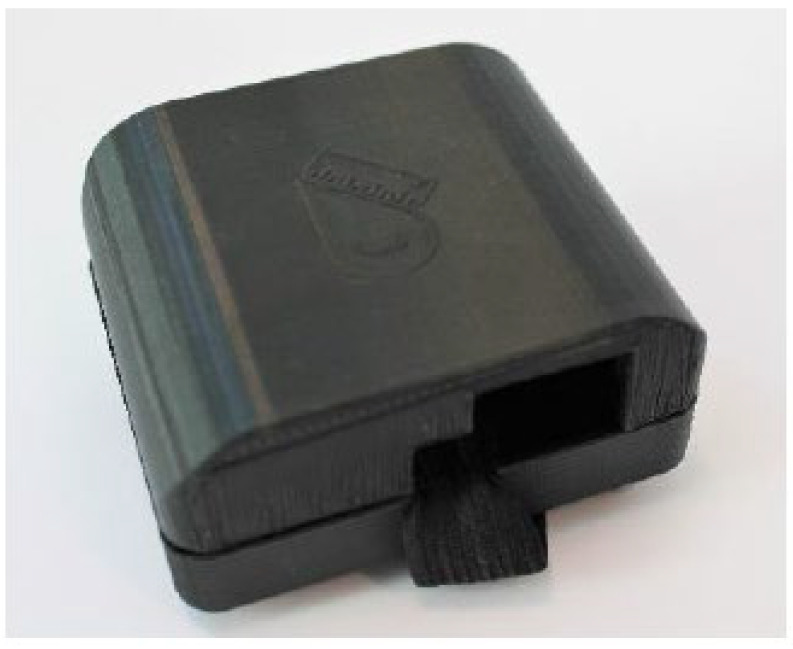
3D-printed enclosure.

**Figure 17 sensors-22-09499-f017:**
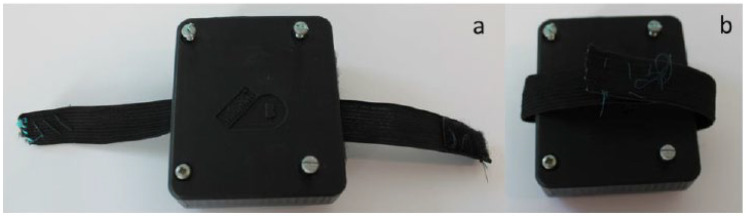
3D-printed enclosure with elastic straps to fit child’s ankle. (**a**) Front, (**b**) Back.

**Figure 18 sensors-22-09499-f018:**
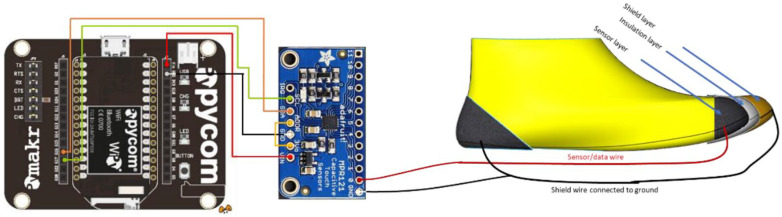
Physical connection from the electronics unit to the shoe sensor unit.

**Figure 19 sensors-22-09499-f019:**
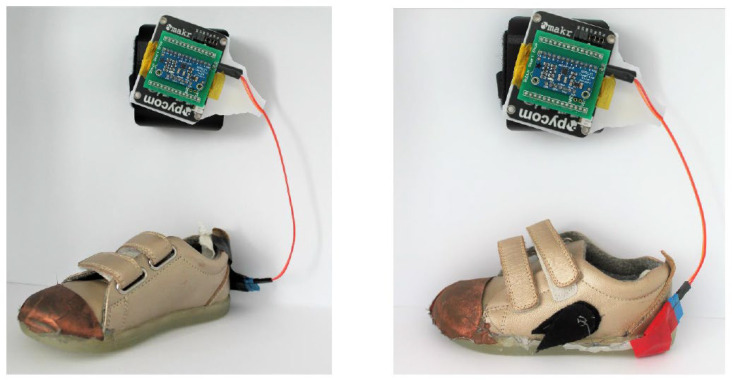
(**Left**) Non-grounded heel sensor connected to Pycom and MPR121, (**Right**) Grounded heel sensor connected to Pycom.

**Figure 20 sensors-22-09499-f020:**
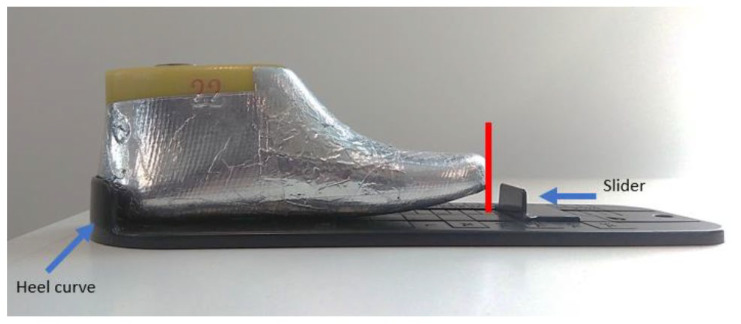
Bobux shoe-size measurement.

**Figure 21 sensors-22-09499-f021:**
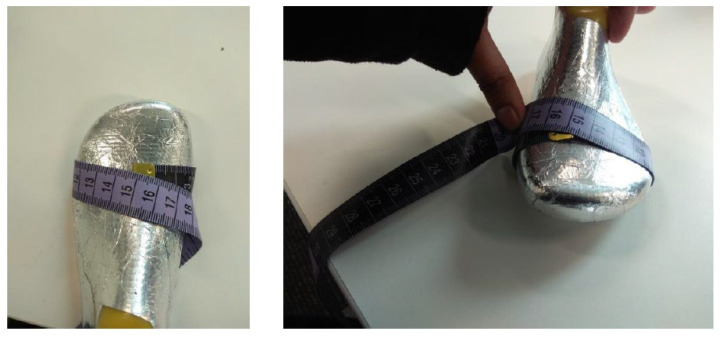
Tape measure wrapped around the widest part of the foot.

**Figure 22 sensors-22-09499-f022:**
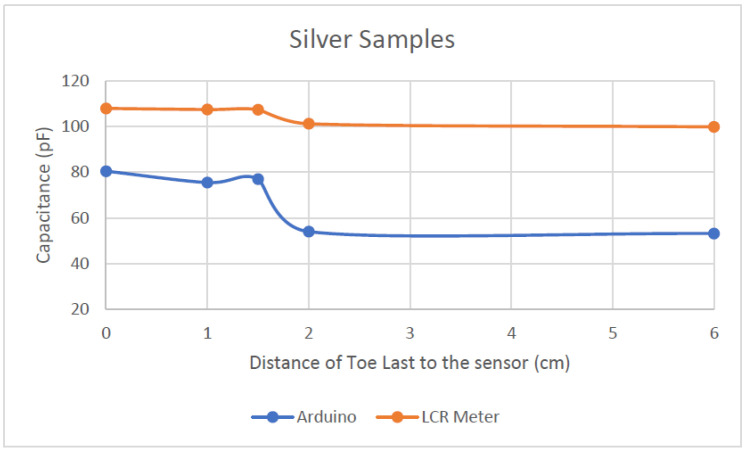
Silver sample capacitance readings comparison between Arduino and LCR meter.

**Figure 23 sensors-22-09499-f023:**
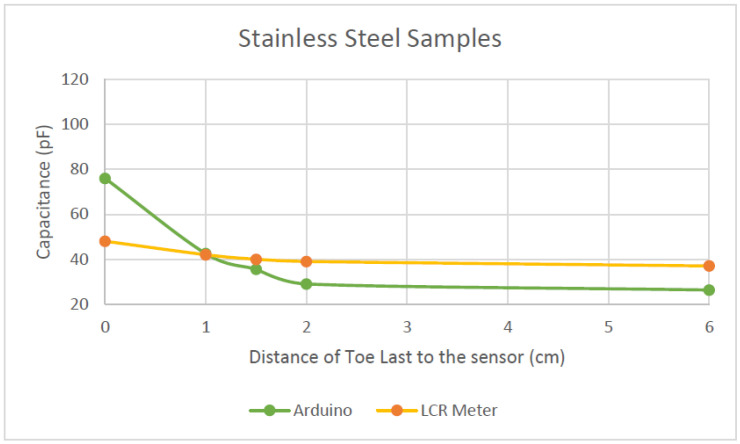
80/20 polyester/stainless steel sample capacitance reading comparison between Arduino and LCR meter.

**Figure 24 sensors-22-09499-f024:**
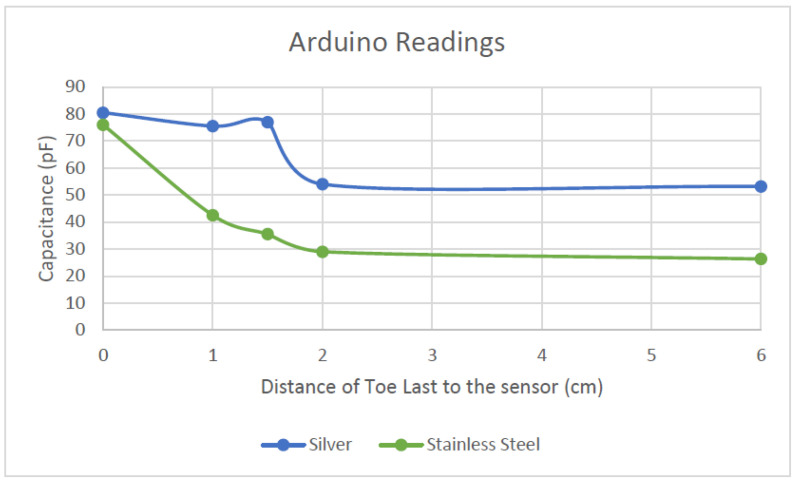
Capacitance readings using Arduino, comparing silver and 80/20 polyester/stainless steel knitted samples.

**Figure 25 sensors-22-09499-f025:**
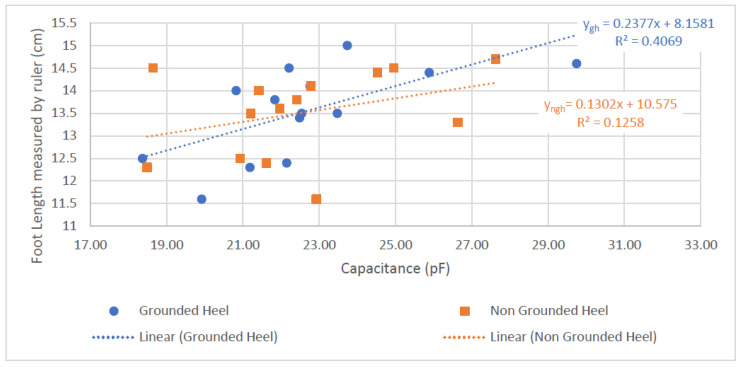
Graph of Foot Length and Capacitance.

**Figure 26 sensors-22-09499-f026:**
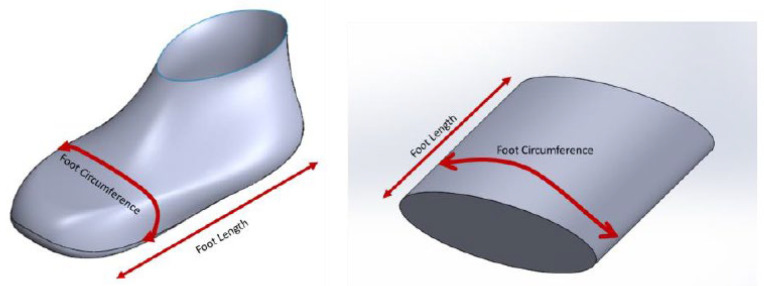
Assumption of the foot as an ellipse shape.

**Figure 27 sensors-22-09499-f027:**
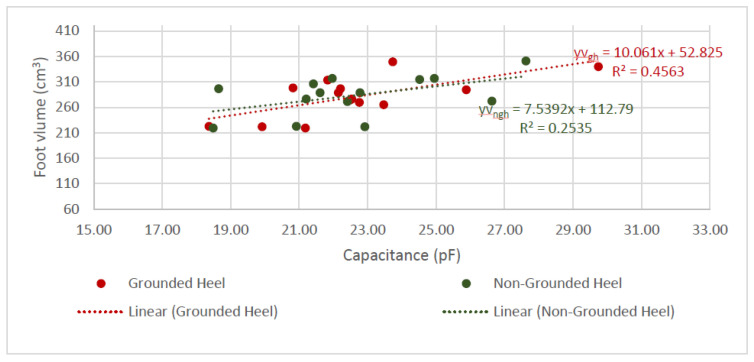
Graph of Foot Volume and Capacitance.

**Table 1 sensors-22-09499-t001:** Capacitance readings from both knitted textile sensors. Readings derived from LCR meter and Arduino.

Shoe Last Inserted	Distance of the Last to the Fabric Sensor (cm)	Capacitance Readings (pF)			
		Silver Samples		Stainless Steel Samples	
		Arduino	LCR Meter	Arduino	LCR Meter
No last	6	53.19	99.92	26.3	37
Size 23	2	54	101.26	29	39
Size 22	1.5	77	107.45	35.5	40
Size 21	1	75.5	105.4	42.5	42
Size 20	0	80.5	108	76	48

**Table 2 sensors-22-09499-t002:** Linear Regression—Statistical Analysis for Capacitance and Foot-Length Relationship.

Linear regression model: *y = mx + cy = equation for both GH and NGH shoes*; *m = gradient of the equation*; *c = y intercept of the model*			
	*m* (*lower CI*, *upper CI*, *SE*)	*c* (*lower CI*, *upper CI*, *SE*)	*p*
*y_ngh_*	0.130 (−0.085, 0.346, 0.0991)	10.57 (5.669, 15.48, 2.252)	0.2135
*y_gh_*	0.238 (0.057, 0.418, 0.0829)	8.158 (4.012, 12.2, 1.889)	0.0141

**Table 3 sensors-22-09499-t003:** Linear Regression—Statistical Analysis for Foot Volume and Capacitance.

Linear regression model *yv = foot volume regression equation = mx + cyv = equation* *for both GH and NGH shoes*; *m = gradient of the equation*; *c = y intercept of the model*			
	*m* (*lower CI*, *upper CI*, *SE*)	*c* (*lower CI*, *upper CI*, *SE*)	*p*
*yv_ngh_*	7.54 (−0.60, 15.68, 3.73)	112.73 (−72.11, 297.6, 84.83)	0.066
*yv_gh_*	10.06 (3.15, 16.97, 3.16)	52.84 (−104.6, 210.3, 72.25)	0.008

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
