# Peer review of "Comprehensive Understanding of Foot Development in Children Using Capacitive Textile Sensors"

_sensors, 2022, doi:10.3390/s22239499_

Round 1

Reviewer 1 Report

However, the main concern of this paper is a lack of fundamental study of capacitive sensors e.g.; Hysteresis of capacitance, repeatability of use, the capacitance changes by pressure and its correlation by foot pressure/sizes/circumferences etc.

Author should rearrange Figures and tables in a logical manner, they are too many (30 figures!), with unnecessary repetition of the same picture (e.g.; 1, 4). Figure 4 may be deleted,  Figure  5, 6, 7;  10 and 11; 13, 14 and 15; 18, 19, 20, 21 and 22; 25, 26, 27 and 28 may combined together separately and align their explanation in text.

Author presented the results in Table 1, Figure 25, 26 and 27, where the capacitance value insignificantly changes by the size or foot distance, how author can guarantee these readable capacitance value changes by the distance 2cm upto 6cm for repeatable uses and what are the standard deviation or error of each measuring point?

Author Response

  1. However, the main concern of this paper is a lack of fundamental study of capacitive sensors e.g.; Hysteresis of capacitance, repeatability of use, the capacitance changes by pressure and its correlation by foot pressure/sizes/circumferences etc.

    The below paragraph has been added on page 2 to shed more light on capacitive sensors:
    Capacitive sensing is based on measuring electrical output through coupling between conductive and dielectric mediums. The change in capacitance detected through these sensors is directly proportional to the changes in pressure applied to the foot. Hysteresis is one of the critical drawbacks of capacitive sensors employing polymers as they tend to lose their water retention properties over time, thus in this study, we made use of the textile sensors to avoid this problem.

  2. Author should rearrange Figures and tables in a logical manner, they are too many (30 figures!), with unnecessary repetition of the same picture (e.g.; 1, 4). Figure 4 may be deleted,  Figure  5, 6, 7;  10 and 11; 13, 14 and 15; 18, 19, 20, 21 and 22; 25, 26, 27 and 28 may combined together separately and align their explanation in text.

    Figure 4 is necessary to show the wiring layout (red and black wires) of the shoe.
    Figures 5 and 7 are combined. Figure 6 was too big to be combined.
    Figures 10 and 11 are combined.
    Figures 14 and 15 are combined
    Other figures are too different or big to be combined

3. Author presented the results in Table 1, Figure 25, 26 and 27, where the capacitance value insignificantly changes by the size or foot distance, how author can guarantee these readable capacitance value changes by the distance 2cm upto 6cm for repeatable uses and what are the standard deviation or error of each measuring point?

The size variations taken in the table are minute (from 20 to 21 to 22 to 23). The LCR meter gives a mixture of both resistance and capacitive values in the table. The changes in the capacitance values with size and pressure changes are considerable.

Reviewer 2 Report

The authors presented a very interesting study of the textile electrodes-based capacitive measurements applied to the children's health aspects of foot growth. 

I have several comments which I hope could improve the paper:

I suggest separating measurement hardware description in the separate section including subsections  2.1.3, 2.1.3.1, 2.2.3

Separately, the section regarding sensors including the discussions about prototypes and their purpose

Please make the figure text font larger

Page 17 line 508. The trend for both Arduino and LCR Meter is not similar and with different sensitivities. Please comment on the difference in the absolute values obtained by capacitance measurement.

— Page 18 line 534. it is possible that this is due to material not regular surface. Please briefly  comment how conductivity of material affects capacitance measurement.

How many measurements were performed at each distance to the sensor?

Figures 28-30 should be placed in the text where they are referenced first.

Author Response

The changes have been made and mentioned in BOLD

The authors presented a very interesting study of the textile electrodes-based capacitive measurements applied to the children's health aspects of foot growth. 

I have several comments which I hope could improve the paper:

— I suggest separating measurement hardware description in the separate section including subsections  2.1.3, 2.1.3.1, 2.2.3

— Separately, the section regarding sensors including the discussions about prototypes and their purpose

These headings have been separated.

— Please make the figure text font larger

Figure text has been made larger

— Page 17 line 508. The trend for both Arduino and LCR Meter is not similar and with different sensitivities. Please comment on the difference in the absolute values obtained by capacitance measurement.

The trend for both Arduino and LCR Meter are similar, however between the 2 cm and 1 cm mark, they have two different slopes. Arduino detected a much larger change on the last centimeter compared to the LCR Meter. Resulting in a much steeper gradient and dramatic change as the foot came much closer to the sensor. 

— Page 18 line 534. it is possible that this is due to material not regular surface. Please briefly  comment how conductivity of material affects capacitance measurement.

From the material evaluation point of view, as observed in Figure 24, the Polyester/Stainless Steel (SS) sample showed a steeper increase in capacitance compared to the Silver Sample. The more conductive material in this area will pick up more capacitance compared to a less conductive one.  

— How many measurements were performed at each distance to the sensor?

At least 5 iterations of 4 readings were made in total

— Figures 28-30 should be placed in the text where they are referenced first.

Agreed, this change has been implemented.